Increasing methane (CH4) emissions and altering rhizosphere microbial diversity in paddy soil by combining Chinese milk vetch and rice straw

Ma Qiaoying
Li Jiwei
Aamer Muhammad
Huang Guoqin hgqjxauhgq@jxau.edu.cn
Research Center on Ecological Sciences, Jiangxi Agricultural University , Nanchang , Jiangxi , China
Joshi Sanket
Electronic publication date: 2020 Aug 3
Publication date: 2020
Volume: 8
Electronic Location ID: e9653
Received 2020 Feb 12; Accepted 2020 Jul 13
Copyright: ©2020 Ma et al.
Copyright year: 2020
Copyright holder: Ma et al.
License: This is an open access article distributed under the terms of the Creative Commons Attribution License, which permits unrestricted use, distribution, reproduction and adaptation in any medium and for any purpose provided that it is properly attributed. For attribution, the original author(s), title, publication source (PeerJ) and either DOI or URL of the article must be cited.
License URL: https://creativecommons.org/licenses/by/4.0/

Keywords: Chinese Milk Vetch, Methane(CH4) emission, 16S rDNA, ITS1, Microbial community diversity

Funding: National Natural Science Foundation of China 41661070 National Key R&D Program of China 2016YFD0300208 Key disciplines (construction) of ecology in the 13th Five–Year Plan of Jiangxi Agricultural University This work was supported by the National Natural Science Foundation of China (41661070), the National Key R&D Program of China (2016YFD0300208), and the Key disciplines (construction) of ecology in the 13th Five–Year Plan of Jiangxi Agricultural University. The funders had no role in study design, data collection and analysis, decision to publish, or preparation of the manuscript

==============================
Background

Chinese milk vetch (Astragalus sinicus L.) can improve paddy soil fertility and ecology through nitrogen fixation, but it can also increase greenhouse gas emissions. Our primary objective was to investigate how Chinese milk vetch, rice straw, and nitrogen fertilization affect the methane and microbial components of the rice rhizosphere.

Methods

We examined the rhizosphere’s methane emissions and microbial abundance and diversity after incorporating Chinese milk vetch and rice straw into paddy soil. We used high-throughput sequencing of the 16s rRNA and ITS1 genes to study changes in the bacterial and fungal communities, respectively. Over the course of our experiment, we applied seven different treatments to the paddy soil: conventional fertilization (the control treatment) for winter fallow crops, three levels of nitrogen in Chinese milk vetch, and three levels of nitrogen in Chinese milk vetch combined with rice straw.

Results

Rice yield and methane emissions increased during cultivation when the soil was treated with Chinese milk vetch with and without added straw. The nitrogen application also affected the methane fluxes. Alpha diversity measurements showed that Chinese milk vetch increased the diversity of the soil fungal community but did not significantly affect the bacterial community. Chinese milk vetch affected the rhizosphere microorganism communities by increasing the number of Methanomicrobia.

Introduction

The excessive use of chemical fertilizers in agricultural production causes serious environmental problems such as soil acidification and aquatic eutrophication, demonstrating the need for more sustainable agricultural practices. In China, the utilization rate of nitrogen fertilizer in main cereal crops is only 30%–50% (Guo et al., 2010). Many studies have explored how to reduce the use of chemical fertilizers while still ensuring the efficiency, yield, and quality of crops treated with nitrogenous fertilizers. Green manure and rice straw management agronomic strategies reduce the use of nitrogen fertilizers, reduce greenhouse gas emissions, and increase yield (Bhattacharyya et al., 2012). As the drawbacks of chemical fertilizers have emerged over recent years, the use of Chinese milk vetch (Astragalus sinicus L.) as a green fertilizer has gained attention. Chinese milk vetch is an important winter fallow crop found in paddy fields in southern China (Zhu et al., 2012). Compared to gramineous green manure, Chinese milk vetch’s lower carbon/nitrogen (C/N) ratio and greenhouse gas emissions make it an ideal winter covering crop (Kim et al., 2013). Long-term rotations of rice and Chinese milk vetch can improve soil fertility, increase soil organic matter, improve soil pH, change the structure of the soil microbial community, and increase the growth of beneficial bacteria (Toda & Uchida, 2017; Zhang et al., 2017). Although green manure and straw return has been shown to increase a rice field’s methane emissions (Zhu et al., 2012; Xu et al., 2017), the amounts used can be adjusted to lower these levels (Lee et al., 2010). Moreover, a previous study found that methane emissions were lower and the impact on rice growth was greater when applying long-term straw returning management than short-term straw additions (Xiao et al., 2019).

Straw and green manure can potentially replace nitrogen fertilizers, but their effect on soil microbial communities is unclear. The rhizosphere is vital for plant nutrition, health, and quality. Rhizosphere soil contains active and diverse microbial communities, and plant residues in the soil are important carbon sources for microorganisms (Chen et al., 2018). Microorganisms decompose organic matter in the soil, cycle soil nutrients, sequester carbon, and maintain soil fertility (Xiao et al., 2018; Berg et al., 2014; Kuypers, Marchant & Kartal, 2018). Oryza sativa, one of the world’s major staple foods (Sharif et al., 2014), has distinct root-associated microbiomes (Edwards et al., 2015). The rhizosphere microbial communities are affected by rice genotype and geographical location, and also transform over the course of the cultivation period (Zhang et al., 2018). The microbiota change dramatically during the vegetative stages, stabilize from the beginning of the reproductive stage, and then undergo relatively minor changes until the ripening stage (Zhang et al., 2018).

It is important to understand the effects of Chinese milk vetch and rice straw on microbial communities in the rice rhizosphere and on methane emissions in rice fields. Rice fields are a significant source of methane emissions, and methane flux is closely related to the abundance of methanogens on rice roots (Pump, Pratscher & Conrad, 2015). The rice roots locally release oxygen and secrete exudates that are rich in organic matrix, creating an ideal habitat for anaerobic and aerobic methanogens (Conrad, 2007). Stubble retention and nitrogen application significantly alter soil microbial community structure and the abundance of carbon turnover microorganisms, thereby increasing CO2 and CH4 flux (Wakelin et al., 2007; Wang et al., 2018).

In the present study, we examined the effects of different fertilizer and rice straw management practices in a Chinese milk vetch-rice rotation system over four years. We studied: (1) changes in soil properties, (2) rice yield and straw production, (3) methane emissions during rice growth, and (4) rhizosphere bacterial and fungal communities and their soil parameter relationships.

Material and Methods

Site description

This field study was part of a long-term experiment in a Dengjiabu original rice seed field (28.22° latitude, 116.85° longitude) in Yujiang District, Yingtan City, Jiangxi Province, China. Lijin Zhang approved soil and plant sample collection on behalf of the Original Seed Farm of Dengjiabu. The experimental site was part of a major Chinese rice production area and planting system. The area has a subtropical humid monsoon climate. Over the past 50 years, the area had an average annual temperature of 17.6 °C, precipitation of 1,788.8 mm, sunshine duration of 1,739.4 h, and elevation of 30 m. The soil texture was 6.8% clay, 70.3% silt, and 23.2% sand. Before treatment, the soil had a pH of 5.59, 34.7 g kg−1 of organic matter content, 1.9 g kg−1 of total nitrogen (TN), 0.5 g kg−1 of total phosphorus, and 22.2 g kg−1 of total potassium.

Experimental design

Field experiment

We conducted the field experiment in a paddy field during winter 2015. We used a split plot design where the main plots had two levels of straw incorporation: 0 kg ha−1 of incorporated straw (V) and 6,000 kg ha−1 of incorporated straw (SV). We based the amount of straw returned to the field on conventional practices and previous studies (Aulakh et al., 2001). After harvesting late-season rice, we immediately removed and weighed the fresh straw from the ground. We cut the straw into approximately 10 cm pieces and returned 6,000 kg ha−1 of straw to the field. Three different levels of nitrogen were applied to the subplots during the Chinese milk vetch growing season: 0 kg ha−1 (a), 15 kg ha−1 (b), and 30 kg ha−1 (c). We used variations on the nitrogen levels and the use of straw to create six treatments: Va, Vb, Vc, SVa, SVb, and SVc. We also established a control treatment (NSV) using winter fallow that was cultivated before the rice (Table 1). We applied each treatment and the control treatment on three plots for a total of 21 plots. Each plot measured 25 m2 (5.0 m × 5.0 m). The fertilizers contained urea (46% N), calcium magnesium phosphate (12% P2O5), and potassium chloride (60% KCl). A typical phosphate and potassium fertilization management system uses 20 kg ha−1 of pure phosphorus and 60 kg ha−1 of pure potassium. Nitrogen fertilizer was applied as a basal, tillering, and booting fertilizer at a 6:3:1 ratio. We applied phosphate fertilizers as basal fertilizers, and potassium fertilizer as a tillering and booting fertilizer at a 7:3 ratio. We applied basal fertilizers the day before rice transplanting, tillering fertilizer 7 days after transplanting seedlings, and heading fertilizer at 20% panicle emergence.

Table 1 Experimental design of the field.

S, straw; V, vetch; a = 0, b = 15, c = 30 kg ha−1 N from vetch; NSV, no straw or vetch. The nitrogen content of the straw was 0.3%, and 6,000 kg is equivalent to 18 kg of nitrogen; therefore, the annual nitrogen application in plots with straw incorporation was reduced by 18 kg ha−1. The N level is the amount of nitrogen applied, and all treatments received the same total amount of nitrogen, 120 kg ha−1.

Treatment	Straw (kg ha−1)	Chinese milk vetch N level (kg ha−1)	Rice N level (kg ha−1)	Winter treatment	
NSV	0	0	120	Fallow	
Va	0	0	120	Chinese milk vetch	
Vb	0	15	105	Chinese milk vetch	
Vc	0	30	90	Chinese milk vetch	
SVa	6000	0	102	Chinese milk vetch	
SVb	6000	15	87	Chinese milk vetch	
SVc	6000	30	72	Chinese milk vetch	

We transplanted the rice variety Yuenuo 06 on April 26, and harvested in early July. We manually harvested each rice plot and measured the aboveground fresh straw and rice yield. We used the same local irrigation management system. Before and after transplanting, we kept the rice submerged in water about three cm deep. Twenty-eight days after transplanting, we implemented a week of mid-term drainage. We irrigated the fields intermittently and did not drain them again until one week before maturity. We sowed the Chinese milk vetch 15 days before the late rice harvest in early October, applying 60 kg ha−1 seed for each plot (excluding the NSV) to the plots. We used the Chinese milk vetch variety Yujiang big leaf seed, which we ploughed into the soil during the bloom stage. At the time of incorporation, fresh Chinese milk vetch showed the following characteristics: 88% water content, 350.52 g kg−1 organic carbon (determined using the dry combustion method), 25.4 g kg−1 total nitrogen (determined using the Kjeldahl digestion method), and a 13.8 C:N ratio. We additionally followed local pesticide and herbicide management practices. Pesticides were applied once during the tillering and booting stages, mainly to prevent borers and sheath blight. Herbicides were applied once during the seedling stage.

Rhizosphere soil sample collection

After 4 years of treatment, we sampled the soils during the early rice ear stage in June 2019. We collected five randomly selected plants from each plot and used a sterile spoon to scrape the soil from the root surface of each plant. We placed the rhizosphere soil in 50 mL sterile centrifuge tubes, added liquid nitrogen, transported it to the laboratory with dry ice, and stored it at −80 °C for microbial analyses. We collected an additional 500 g of bulk soil from each plot using the five-point snake sampling method, mixed the samples together (Nanjing Soil Research Institute CA of S, 1978), placed them in a self-sealing bag, and transported the samples to the laboratory. This soil was air-dried for physicochemical analysis.

We measured soil pH using a 1:5 soil-water ratio extraction method and a glass electrode (E-301-CF; LEICI). We used an automated flow injection analyzer (FIA, Lachat Instruments, Loveland, CO, USA) to measure nitrate and ammonium nitrogen levels. Total organic carbon (TC) was measured using dry combustion with a macro elemental analyzer (Vario MAX C/N; Elementar Analysensysteme, Hanau, Germany). We measured soil TN using the Kjeldahl digestion method (Henry et al., 2006). Available phosphorus (AP) was measured by extracting the samples in 0.5 M NaHCO3. We measured available potassium (AK) using flame photometry (Clark et al., 1998), and microbial biomass carbon (MBC) and microbial biomass nitrogen (MBN) using chloroform fumigation and K2SO4 extraction, respectively (Kudeyarov & Jenkinson, 1976; Shen, Pruden & Jenkinson, 1984). We measured soil activated organic carbon (AOC) using the 333 mmol L−1 potassium permanganate oxidation method (Blair, Lefroy & Lisle, 1995). We measured water content and bulk density using the methods reported by Avnimelech et al. (2001).

Soil genomic DNA extraction and HiSeq sequencing

Total genomic DNA was extracted from rice rhizosphere soil using a FastDNA SPIN Kit for Soil (MP Biomedicals, Solon, OH, USA) according to the manufacturer’s instructions. To examine bacterial community diversity, we amplified the V4 region of the 16s rDNA gene using the primers 515F (5′-GTGCCAGCMGCCGCGGTAA-3′) and 806R (5′-GGACTACHVGGGTWTCTAAT-3′) (Zhang et al., 2017). To examine fungal community diversity, we amplified the ITS1 gene region using the primers ITS1F (5′-CTTGGTCATTTAGAGGAAGTAA-3′) and ITS1R (5′-GCTGCGTTCTTCATCGATGC-3′) (Schmidt et al., 2019). We then barcoded the forward and reverse primers. We used a 30 ng genomic DNA sample and the corresponding fusion primers to configure the PCR reaction system (Gene Amp PCR-System® 9700) (Zhang et al., 2017). Agencourt AMPure XP magnetic beads (Beckman Coulter, CA, USA) were used to purify the PCR amplification products, which were then dissolved in elution buffer, labeled, and used to construct the library. We cleaned the final amplicon libraries twice using the Agencourt AMPure XP Kit (Beckman Coulter GmbH, USA) and sequenced them on a BGI HiSeq2500 platform (BGI Huada, Shenzhen, China). We submitted the raw reads to the NCBI Sequence Read Archive database under Bioproject PRJNA587576, ITS1 sample accession numbers SAMN13671326 to SAMN13671346. and 16s rDNA sample accession numbers SAMN13195521 to SAMN13195541.

Pot experiment

We conducted a parallel field incubation pot experiment on a Jiangxi Agricultural University experimental field (28.77° latitude, 115.84° longitude). In March 2019, we collected soil samples from Dengjiabu Orientation experimental field plots. We incubated four pots for each treatment and merged the results. We placed the plastic pots on the surface of the soil and used the same treatment conditions as the field experiment. We collected soil samples (0 to 20 cm) using a five-point sampling method for each treatment. Each sample was mixed, air-dried, passed through a six mm sieve, and the stones were removed.

We filled the plastic pots (25 cm height, 21 cm bottom diameter, 25 cm top diameter) with 7.0 kg of dry soil to approximately the depth of field cultivation (20 cm). The freshly harvested Chinese milk vetch (100 g) were cut into three cm segments, incorporated into the pots (except in the NSV group), and incubated with 2–3 cm of water. We replenished the water every day after measuring the height of the water surface. A nylon mesh bag (37 µm mesh size (400 mesh), 10 cm diameter, 15 cm height) was placed in the center of each pot at a depth of 15 cm, dividing the pot into two soil compartments: a central root compartment and a surrounding non-root compartment. We inserted an automatic temperature and humidity meter (ZDR-20, Zeda Instruments, Hangzhou, China) into each pot to record the temperature and humidity every half hour. On April 25, 2019, two healthy early rice seedlings were transplanted into each pot’s root bag. We applied N, P, and K fertilizer in each pot according to their corresponding field treatment. A 2–3 cm layer of water was maintained during rice cultivation. After transplanting the rice, we measured CH4 emissions using the static closed-chamber method at 7 day intervals (Zou et al., 2005; Jia, Cai & Tsuruta, 2006). We analyzed gas concentration using a gas chromatograph (Agilent 7890B, CA, USA). The CH4 emissions were calculated using previously described methods (Afreh et al., 2018; Xiao et al., 2019). CH4 flux was calculated using the following equation: F=ρ×V∕A×ΔC∕Δt×273∕273+T

where F is the emission flux rate of CH4 (mg m−2 h−1), ρ is the gas density of CH4 under a standard state (mg m−3), V is the chamber volume (cm3), A is the soil area, ΔC is the concentration of CH4 produced during an hour sealed, Δt is the time between sample collection (h), and T is the mean chamber temperature (°C).

The experiment ended 11 weeks after rice transplantation. During this period, approximately 90% of seasonal CH4 emissions were generated (Xiao et al., 2019). At the end of the pot experiment, we harvested the plants and measured the above- and below-ground biomass. We oven-dried rice plants at 70 °C for 24 h to reach a constant weight. To determine physicochemical properties, we air-dried the remaining soil after harvesting using the same method as the field experiment.

Microbial diversity analysis

We used USEARCH software (v10.0.240, Robert Edgar) to cluster the stitched tags into amplicon sequence variants (ASVs). We also used USEARCH for dereplication, and unoise3 to denoise, predict biological sequences, and filter chimeras. We used the 16S database Greengene (V201305) for bacteria and the ITS fungus database UNITE (Version 6 20140910) for fungi. We filtered the annotated results to remove ASVs without unannotated results. We used USEARCH to calculate the alpha diversity index, then calculated the richness change of the dilution process and rarefaction from 1% to 100% in abundance (observed OTUs). The constrained canonical correspondence analysis (CCA) profile used a Bray-Curtis distance matrix (Hannula et al., 2017) and R package amplicon (Liu, 2019). We used a partial Mantel test to evaluate the relationships between soil microbial community richness and environmental factors. We deposited these raw reads in the NCBI Sequence Read Archive database.

We used R (version 3.6.3) (R Core Team, 2020) for data analyses and plotting. We used Bartlett’s test of homogeneity, and applied Fisher’s Least Significant Difference (LSD) test for multiple comparisons. Related packages included agricolae (Felipe, 2020), vegan....... (Oksanen et al., 2019), and ggplot2 (Wickham, 2016). We used the analysis of similarities (ANOSIM) function to find similarities across the microbial community composition data.

Table 2 Effect of Chinese milk vetch and rice straw on soil properties.

NSV, Va, Vb, Vc, SVa, SVb, SVc was treatment as write in text. Values (mean standard deviation) indicate the absolute amount of each characteristic. Different letters in a row indicate a significant difference at P < 0.05 (n = 3). Analysis of variance (ANOVA) was performed. TC, total carbon; TN, total nitrogen; AOC, activated organic carbon; MBC, microbial biomass carbon; MBN, microbial biomass nitrogen; NO3-N, Nitrate Nitrogen; NH4-N, ammonium nitrogen; AN, alkali-hydrolysable nitrogen; AK, available potassium; AP, available phosphorus; BD, bulk density; WFPS%, water filled pore space.

	NSV	Va	Vb	Vc	SVa	SVb	SVc	
TC ( g kg−1 )	16.5 ± 0.72b	17.73 ± 1.1ab	17.5 ± 1.35ab	16.07 ± 1.7b	19.67 ± 1.4a	18.5 ± 0.78ab	18.7 ± 3.2ab	
TN ( g kg−1 )	1.76 ± 0.16a	1.88 ± 0.25a	1.83 ± 0.14a	1.77 ± 0.18a	1.79 ± 0.2a	1.81 ± 0.18a	1.95 ± 0.07a	
AOC (mg kg−1 )	424.36 ± 86.77a	533.62 ± 182.73a	505.08 ± 181.76a	488.26 ± 163a	457.34 ± 41.32a	595.46 ± 132.44a	558.33 ± 245.78a	
MBC (mg kg−1 )	363.28 ± 80.67b	357.58 ± 159.23b	538.73 ± 103.48ab	332.58 ± 84.86b	442.92 ± 66.99ab	606.58 ± 209.01a	465.81 ± 95.23ab	
MBN (mg kg−1 )	30.55 ± 12.14b	45.02 ± 10.04ab	48.24 ± 12.76a	46.63 ± 7.37a	45.02 ± 2.79ab	54.67 ± 2.79a	40.2 ± 7.37ab	
NH4-N(mg kg−1 )	1.69 ± 0.59bc	2.64 ± 0.92ab	1.81 ± 0.58bc	1.8 ± 0.79bc	1.31 ± 0.3c	3.4 ± 0.56a	1.51 ± 0.45c	
NO3-N(mg kg−1 )	0.12 ± 0.12a	0.11 ± 0.05a	0.13 ± 0.07a	0.09 ± 0.06a	0.15 ± 0.08a	0.1 ± 0a	0.11 ± 0.08a	
AN (mg kg−1 )	491.87 ± 51.6a	557.9 ± 84.28a	569.33 ± 65.7a	470.87 ± 80.56a	500.27 ± 61.01a	495.6 ± 63.84a	545.53 ± 55.73a	
AK (mg kg−1 )	45.09 ± 4.73c	58.84 ± 7.75abc	64.71 ± 38.56bc	49.65 ± 17.42abc	65.84 ± 8.84ab	69.18 ± 25.28a	56.1 ± 10.37abc	
AP (mg kg−1 )	10.52 ± 6.09ab	7.12 ± 0.32b	10.1 ± 3.09ab	11.13 ± 2.67ab	10.01 ± 2.26ab	13.13 ± 4.68a	10.2 ± 0.29ab	
pH(H2O,1:5)	5.11 ± 0.05a	5.12 ± 0.11a	5.06 ± 0.02a	5.13 ± 0.02a	5.1 ± 0.07a	5.14 ± 0.05a	5.07 ± 0.08a	
BD (g cm−3 )	1.19 ± 0.03a	1.15 ± 0.06abc	1.06 ± 0.07bc	1.18 ± 0.06ab	1.06 ± 0.09bc	1.07 ± 0.12abc	1.04 ± 0.02c	
WFPS( %)	65.24 ± 2.22a	62.66 ± 3.16a	57.94 ± 6.28a	64.02 ± 7.34a	58.28 ± 6.18a	59.05 ± 8.28a	59.3 ± 1.83a	

Results

Physicochemical properties of paddy soil

During the heading stage in June, we collected soil to analyze the mid-term effects of rotation. When compared to the NSV group, the other treatments changed most of the soil’s properties (Table 2). The TC levels of the SVa treatment group were significantly higher than those of the NSV treatment. TN, AOC, MBC, MBN, and AP levels from the field experiment were also higher. Incorporating Chinese milk vetch and straw into the field had no significant effects on AN, nitrate nitrogen, ammonium nitrogen, and AP levels. The soil bulk density after Chinese milk vetch and straw incorporation were significantly lower than that of the NSV group (Table 2).

Figure 1 Effects of different levels of nitrogen from Chinese milk vetch with and without straw incorporation into soil on (A) rice and (B) straw yield.

S, straw; V, vetch; a = 0, b = 15, c = 30 kg ha−1 N from vetch; NSV, no straw and vetch. Different letters represent significant differences (P < 0.05) (n = 3).

Figure 2 Effects of different levels of nitrogen from Chinese milk vetch with and without straw incorporated into soil on methane emissions.

S, straw; V, vetch; a = 0, b = 15, c = 30 kg ha−1 N from vetch; NSV, no straw and vetch. Different letters represent significant differences (P < 0.05). Error bars represent the standard deviation (n = 3).

Rice yield and greenhouse gases

In the field study, treatments significantly (P < 0.05) increased the rice yield compared to the NSV group. The Va treatment produced the highest rice yield, 5,306.67 kg ha−1, which was 1.27 times higher than the NSV group. This was followed by the SVa treatment. The yields from SVa > SVb > SVc, and Va > Vb and Vc. The straw yields showed a similar trend (Figs. 1A and 1B). In the pot experiment, the biomass dry weight of the shoot and root treated with Chinese milk vetch and straw was higher than that of the NSV treatment (Figs. S1A and S1B). The NSV group’s average CH4 emission of 2.56 mg m−2 h−1 was significantly lower than that of the other treatments. The SVa treatment produced the highest average CH4 emissions. The mean CH4 flux was directly proportional to the amount of nitrogen fertilizer applied in the Chinese milk vetch and rice straw treatments: SVa> SVb> SVc, and Va> Vb> Vc (Fig. 2).

Rhizosphere microbial community

We used the 16s rRNA and ITS1 gene copy numbers to determine the amount and diversity of microorganisms. After quality filtering the 16s rDNA and ITS1 gene sequences, we obtained 1.097 million sequences and 1.066 million sequences, respectively. The rarefaction curves had flat ends, indicating that the amount of sequencing data was large enough to characterize the samples (Figs. S2A and S2B). The Chinese milk vetch and straw did not have a significant effect on the number of ASVs in the bacterial communities except in the SVb group where the number of bacterial ASVs was significantly lower. Additionally, the SVb group had the lowest Chao1 and Shannon indices (Table 3A). The Shannon fungal indices in the NSV group were lower than in the other treatments (Table 3B). The constrained CCA profile using a Bray-Curtis distance matrix showed the differences across the different treatment samples (Figs. S2C and S2D).

Table 3 Effects of Chinese milk vetch and straw on bacterial diversity and fungi diversity in rhizosphere soil.

(A) Bcteria; (B) Fungi. Diversity was based on a 16S rDNA (bacteria) and ITS1 (fungi) gene sequence assignment data set with a 97% sequence similarity threshold. Three indexes of diversity were examined: Chao1, Shannon’s, and Simpson’s. Values (mean ± standard deviation) were tested in Analysis of variance (ANOVA). Data within the same column followed by the same lowercase letters are not significantly different at P < 0.05 ( n = 3).

	NSV	Va	Vb	Vc	SVa	SVb	SVc	
(A)	
Tag number	51923 ± 1235a	52230 ± 681a	52029 ± 569a	52334 ± 8667a	51756 ± 957a	52546.7 ± 260.1a	52755 ± 1119a	
ASV number	4389 ± 107a	4438 ± 128a	4515 ± 117a	4295 ± 85a	4509 ± 95a	4036 ± 272b	4450 ± 29a	
Chao1	5454.7 ± 124.5a	5387.0 ± 81.5a	5528.5 ± 140.4a	5335.4 ± 137ab	5575.4 ± 102.8a	5029.5 ± 355.8b	5496.6 ± 135.5a	
shannon	6.91 ± 0.09ab	6.99 ± 0.1a	7.01 ± 0.03a	6.91 ± 0.04ab	7.00 ± 0.07a	6.76 ± 0.18b	6.98 ± 0.05a	
simpson	0.003 ± 0.001ab	0.003 ± 0.001ab	0.003 ± 0.000b	0.004 ± 0.001ab	0.003 ± 0.000b	0.004 ± 0.001a	0.003 ± 0.004b	
(B)	
Tag number	51050 ± 555a	50019 ± 495a	50941 ± 628a	51135 ± 59a	50790 ± 494a	50773 ± 1246a	50671 ± 589a	
ASV number	496 ± 105ab	644 ± 87a	567 ± 43ab	465 ± 41b	524 ± 27ab	549 ± 186ab	535 ± 42ab	
Chao1	505.39 ± 109.3ab	669.80 ± 90.6a	587.33 ± 44.9ab	493.57 ± 45.7b	543.62 ± 33.0ab	570.48 ± 203.8ab	554.52 ± 55.5ab	
shannon	3.58 ± 0.42b	4.19 ± 0.27ab	4.37 ± 0.29a	4.02 ± 0.04ab	3.95 ± 0.61ab	3.80 ± 0.84ab	4.19 ± 0.10ab	
simpson	0.111 ± 0.078a	0.062 ± 0.022a	0.037 ± 0.012a	0.051 ± 0.01a	0.073 ± 0.061a	0.096 ± 0.085a	0.044 ± 0.004a	

Microbial composition of rhizosphere soil

We found affiliations between the 16s rDNA gene sequences and 35 phyla. The most abundant phylum was Proteobacteria. The abundance of Methanomicrobia in the NSV treatment was significantly lower than in the other treatments except for SVb at the class level (Fig. 3A). Methanomicrobia is comprised of three orders: Methanocellales, Methanomicrobiales, and Methanosarcinales. The abundance of Methanosarcinales was greater than the other two orders. We found a significantly greater abundance of Mehanocellales in the Va treatment than in the other treatments, and a greater abundance of Methanomicrobiales and Methanosarcinales in the SVa treatment. We found a greater amount of Clostridiales in the Chinese milk vetch and straw treatment groups than in the NSV group (Data S1).

Figure 3 Microbial abundance analysis.

(A) Stacked plot of bacterial class based on the 16s rDNA gene. (B) Stacked plot of fungal class based on the ITS1 gene. Unclassified are unknown new species pooled. S, straw; V, vetch; a = 0, b = 15, c = 30 kg ha−1 N from vetch; NSV, no straw and vetch.

We detected a total of 12 phyla in the ITS1 sequences. Ascomycota was the dominant group in the NSV treatment, where it was significantly more abundant than in the other treatments. Other dominant phyla included Mortierellomycota, Chytridiomycota, Basidiomycota, Rozellomycota, and Glomeromycota. At the class level, the abundance of Dothideomycetes and Sordariomycetes in the NSV group was higher than in other treatments. The abundance of an unclassified species belonging to Glomeromycota was lower in the NSV group than in the other treatments (Fig. 3B).

Relationship between soil parameters and rhizosphere microorganisms

We used a partial Mantel test to analyze the relationship between rhizosphere microbial community diversity and environmental factors (Figs. S3 and S4). Although we found no significant correlations between rhizosphere microorganisms and environmental factors, we observed different trends. Bacterial community diversity had a negative correlation with TN, MBC, MBN, NH4-N, AK, and AP levels, and a positive correlation with bulk density (BD), water-filled pore space (WFPS), and CH4 emissions. Fungal community diversity had a negative correlation with TN, AOC, and AP levels, and a positive correlation with water-soluble organic carbon (WSOC), MBC, MBN, NH4-N, NO3-N, pH, and CH4 levels.

Discussion

Effects of Chinese milk vetch and straw on rice yield and CH4 emissions

Chinese milk vetch is a viable option for a sustainable winter cover crop because it improves both soil fertility and crop yield (Samarajeewa et al., 2005). Chinese milk vetch can also increase soil pH (Zhang et al., 2017), although this effect was not observed in our study. This may be due to the texture of the soil, and we will continue to observe this indicator in future experiments (Table 2). Although Chinese milk vetch has been shown to significantly increase CH4 emissions in previous studies (Lee et al., 2010; Hwang et al., 2017; Xu et al., 2017), it may be more suitable than other winter green manure types, such as ryegrass and legumes, due to its low C/N ratio and comparatively lower CH4 emissions (Kim et al., 2013).

CH4 flux in paddy fields is closely related to soil organic matter content and anaerobic conditions (Liu et al., 2014). Adding Chinese milk vetch to the field at the flowering stage rapidly increased the amount of easily decomposable organic matter in the soil. The addition of organic matter was conducive to the growth of methanogenic bacteria, resulting in increased CH4 emissions (Conrad & Klose, 2006). It may be possible to reduce CH4 flux by incorporating ditch-buried concentrated straw into the field (Liang-yu et al., 2013). 120 kg ha−1 of nitrogen was applied during the rice and Chinese milk vetch growth periods for each treatment (Table 1). We found that after returning rice straw to the field, less nitrogen needed to be applied during the rice cultivation period, which decreased the CH4 efflux (Fig. 2).

Effect of Chinese milk vetch rotation on rhizosphere microorganisms

Soil microbial processes and structure are indicators of soil quality (Turco, Kennedy & Jawson, 1994). Long-term legume cultivation changes the structure and diversity of soil microbial communities (Toda & Uchida, 2017), and some rhizobia can interfere with root development (Garrido-Oter et al., 2018) and substantially change the structure and function of soil microbial communities in the rhizosphere (Zhou et al., 2018).

Straw incorporation is an agricultural policy that has been prioritized by the Chinese government over recent years (Zhao et al., 2018). However, as the amount of straw return increases, the abundance of methanogenic bacteria in rice fields also significantly increases (Asakawa et al., 1998; Conrad & Klose, 2006). In the current study, we found that the abundance of Methanomicrobia in the NSV group was lower than in the other Chinese milk vetch and rice straw treatment groups, producing high methane emissions (Fig. 3A). After long-term rice-Chinese milk vetch rotation, we determined that growth-promoting rhizobacteria Acineobacter and Pseudomonas were the preponderant groups in the rice rhizosphere (Zhang et al., 2017).

The fungi alpha diversity in the NSV group was lower than in the Chinese milk vetch-incorporated treatments (Table 3). Adding appropriate amounts of organic matter can significantly increase the diversity of arbuscular mycorrhizal fungi (AMF) species in farmland soil (Gosling et al., 2010). We found that the abundance of an unclassified species belonging to Glomeromycota was higher in Chinese milk vetch-incorporated treatments than in the NSV group (Fig. 3B). This suggests that Chinese milk vetch may be beneficial to AMF accumulation in paddy soil. Additionally, fungi may have a greater effect on soil carbon cycling than bacteria in acidic soil (Xiao et al., 2018).

Our research on rice field microorganisms was limited to the rhizosphere and thus did not reflect the entire field. It was notable that the organic matter input effects on microbial diversity were relatively minor and that we detected no correlations with soil parameters. Our next objectives are to study the changes in the entire rice field after incorporating Chinese milk vetch and straw, the nutritional interactions between fungi and bacteria, how those interactions change the energy flow in the soil food web (Rudnick, Van Veen & de Boer, 2015), the functional characteristics of different rhizosphere microorganisms in their ecological sites, and the interactions between bacteria and fungi in regulating soil nutrient cycling and crop nutrient uptake to guide agricultural production.

Conclusions

In the present study, we found that incorporating Chinese milk vetch with and without straw in paddy soil produced greater rice yield and CH4 flux than the NSV treatment group. The effects of Chinese milk vetch on CH4 emissions correlated with nitrogen management, and CH4 flux during rice cultivation was directly proportional to nitrogen application. Chinese milk vetch affected the rhizosphere microorganism communities by increasing the relative number of methanogens. In terms of rice yield, soil parameters, and methane emissions, we determined that the SVb treatment (15 kg ha−1 of nitrogen fertilizer allocated for Chinese milk vetch cultivation and 105 kg ha−1 for rice cultivation) was the most effective.

Supplemental Information

Supplemental Information 1 Effects of different levels of nitrogen from Chinese milk vetch with and without straw incorporation into soil on shoot biomass and root biomass

S, straw; V, vetch; a = 0, b = 15, c = 30 kg ha−1 N from vetch; NSV, no straw and vetch. Different letters represent significant differences (P < 0.05) (n = 3).

Click here for additional data file.

Supplemental Information 2 Microbial diversity analysis.

(A) Bacterial 16s rDNA gene rarefaction curves for the different treatments, with standard errors. (B) Fungal ITS1 gene rarefaction curves for the different treatments, with standard errors. (C) 16s rDNA gene constrained CCA. (D) ITS1 gene constrained CCA.

Click here for additional data file.

Supplemental Information 3 Mantel correlograms of the relations between abundance of the bacterial community and soil parameters

S, straw; V, vetch; a = 0, b = 15, c = 30 kg ha−1 N from vetch; NSV, no straw and vetch.

Click here for additional data file.

Supplemental Information 4 Mantel correlograms of the relations between abundance of the fungal community and soil parameters

S, straw; V, vetch; a = 0, b = 15, c = 30 kg ha−1 N from vetch; NSV, no straw and vetch.

Click here for additional data file.

Supplemental Information 5 The order level of bacteria dataset.

NSV, Va, Vb, Vc, SVa, SVb, SVc was treatment as write in text.

Click here for additional data file.

Supplemental Information 6 Raw data of Fig. 1 effects of different levels of nitrogen from Chinese milk vetch with and without straw incorporation into soil on above- and belowground biomass of rice

Click here for additional data file.

Supplemental Information 7 Raw data of Fig. 2 effects of different levels of nitrogen from Chinese milk vetch with and withoutstraw incorporation into soil on methane emissions

Click here for additional data file.

Supplemental Information 8 Raw data of Table 2 effect of long-term winter planting of green manure and straw incorporation on soil properties

Click here for additional data file.

Supplemental Information 9 Raw data of Table 3

Diversity was based on a 16S rRNA (bacteria) and ITS1 (fungi) gene sequence assignment data set with a 97% sequence similarity threshold

Click here for additional data file.

Additional Information and Declarations

Competing Interests

Author Contributions

Field Study Permissions

Data Availability

The authors declare there are no competing interests.

Qiaoying Ma and Jiwei Li conceived and designed the experiments, performed the experiments, analyzed the data, prepared figures and/or tables, authored or reviewed drafts of the paper, and approved the final draft.

Muhammad Aamer analyzed the data, prepared figures and/or tables, and approved the final draft.

Guoqin Huang conceived and designed the experiments, analyzed the data, authored or reviewed drafts of the paper, and approved the final draft.

The following information was supplied relating to field study approvals (i.e., approving body and any reference numbers):

Lijin Zhang approved soil and plant sample collection on behalf of the Original Seed Farm of Dengjiabu.

The following information was supplied regarding data availability:

The 16S rDNA sequences are available at SRA: SAMN13195521 to SAMN13195541. The ITS1 sequences are available at SRA: SAMN13671326 to SAMN13671346.

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
