# Peer review of "Increasing methane (CH4) emissions and altering rhizosphere microbial diversity in paddy soil by combining Chinese milk vetch and rice straw"

_PeerJ, doi:10.7717/peerj.9653_

## Round 0.1 · original submission · Major Revisions

The topic of the manuscript is quite relevant, with increasing awareness of issues related to usage of chemical fertilizers and associated environmental problems. Greener alternatives, as reported in current study could provide basis for environmental friendly substitutes. However, as pointed by reviewers, manuscript needs substantial corrections, some specific suggestions are:

Overall language needs corrections. Such as, refrain starting a sentence with numbers (L26); L38-40, L61-63 and other instances are either incomplete or needs corrections; ‘16s rRNA’ should be corrected at several places, etc.

In-text reference should be changed as per journal style - AI, (for e.g., Kuypers, Marchant & Kartal, 2018, and many instances in the manuscript).

Some of the sections/information need further data (especially materials and methods section) and to be re-written for clarity.
Statistical analysis needs more clarification and details; discussion section needs to be re-written, considering all the results.

The sequencing data does not have any accession number, it should be deposited to some database and the information should be provided.

Please see detailed suggestions and comments by reviewers.

·

Basic reporting

There are a few sections that need to be rewritten for clarity and consistency.

• What is green manure? Define in introduction (first use line 38). try to be consistent with usage of terminology throughout manuscript.
• Line 40 should read ‘… is also unclear’
• five-point sampling method … citation needed (line 122)
• missing quality control steps in methods, including parameters used (line 179)
• What R packages are you using to perform CCA, Bray-Curtis, etc. calculations? (line 179)
• Provide citations for the R packages when included
• Were there any post processing of the data? Counts turned to relative abundance? This information should also be present in the methods.
• ‘If’ should be ‘if’ (line 206)
• sentence on line 206, “These results indicated ...” should be rewritten for clarity.
• Dilution curve should be rarefaction curve (line 225)
• Line 258 is better written: ‘...because soil fertility and crop yield are improved...”

Figure 2. It may be better to show all data points on your plot to more intuitively visualize the variation in the data. I would recommend boxplots here with your individual data points overlayed.

Experimental design

In general, describing your methods in more detail, including those used in data analysis, would help the reader understand in more detail what was done with the data without assumptions.

Validity of the findings

There is some inconsistency or a lack of clarity in the presentation of the results in the discussion. Please see below. In general, I think elaborating further in the discussion would be beneficial as there are many results that are not discussed.

• It is possible to perform an an analysis of similarity (anosim R package) on the microbial community composition data (Fig 3C/D) to test if the groups had significantly different compositions. This would strengthen this area. Additionally, perMANOVA (adonis2 package) would also work here.


• line 220 .. Please comment on the lack of differences between experimental treatment. It would seem the statement is correct relative to the control, however additional nitrogen in the form of Chinese milk vetch seems to reduce the methane emissions in the presence of straw.
• The section ‘Effect of green manure and straw on rice yield and CH4’ seems to contradict the results of the present study. The results of this study indicate that additions of straw and Chinese milk vetch increase methane emission (figure 2), which contradict the statements throughout this section. Elaborating further on why these results contradicts expectations from the literature should be included.
• It is a recurring statement that the short term application of the Chinese milk vetch may have contributed to the observed results of the present study. A comment on what the mechanism is that contributes to a difference over longer application periods would be beneficial to the reader.

Reviewer 2 ·

Basic reporting

Rice is one of the most important staple foods worldwide, yet it is known that most of rice cultivation, carried out under flooded conditions results in high amounts of methane emitted during waterlogged conditions. Only a fraction of applied chemical fertilizer is utilized by plant and the microbes, whereas the rest leaches into the groundwater. Finding a green alternative, which does not result in large yield loss of rice is scientifically as well as socio-economically important matter.
Current manuscript investigates an important environmental problem from the microbiological diversity perspective. Most of the methane is produced by methanogenic archaea, anaerobes known to thrive under waterlogged conditions. Furthermore some aerobic methanotrophs are dependent on nitrate-such as ANME-2d archaea and nitrite, such as M. oxyfera bacteria, as electron acceptor. The healthiness of soil is dependent on the microbial community of the soil. Novelty of this manuscript lays in investigating the fungal community alongside that of the bacterial. Fungi as the ultimate degraders of complex organic matter, supply the soil with bioavailable carbon, yet have not been addressed as often as bacteria even though they play a key role in the carbon cycle.

The written english language would benefit from a second check as at times verbs are missing in the sentences. It is clearly structured, although in the introduction the link between methane and microbes is not discussed, although majority of methane under waterlogged conditions is produced by methanogenic archaea. Nor are important microbial groups considered.
The sequencing data referred to at lines 185-186 does not have an accession number under which it was deposited.
The manuscript would be easier to read if treatments wre referred to with always same terms, currently f.e. green manure and effcts f chinese milk vetch are used interchangably.
The figure legends should be rechecked. On figure 1, the error bars lean to different sides.

Experimental design

Research question is defined, and surely authors have done a lot of experimental work as well as sequenced the data and checked chemical properties of the soil.
The section of materials and methods needs some more clarifying as authors refer to local practices which are unknown for the reader or us terms as reasonable amount of data. The way statistical analysis was conducted in R needs more clarification and details.

Validity of the findings

Despite of a lot of work being carried out, the findings are not very well presented, analysed or in depth described. For methane emissions, it is known that once fields become flooded and rice is cultivated, methane emissions do increase. Rice straw addition does increase methane emissions as well. What is the role of the vedge addition remains unclear. The manuscript needs a bit more structuring as it is not clear how the pot and field experiment on different years and soils can be compared. One of the large analysis methods relies on 16S rRNA gene to address the microbial community composition. For 16S one can go down to atleast to order and family level. Authors have a lot o sequencing data, yet results contain a very small section on most abundant phyla with the second one remaining unassigned. This should be rechecked. As these results are presented in context of methane, the methanotrophic bacteria found in this dataset deserves a closer look.
In the discussion section, a lot of comparable research has been carried out, as paddy fields are well studied ecosystem, though it is no referenced and should be revised carefully.

Additional comments

The current manuscript has potential as it adresses an important topic of rice as staple food, grenhouse gas emissoins and unerlaying microbial community in the rhizosphere.
In the introduction covering the aspect of methane and microbes should be included.
In the section of materials and methods, to use only one term to refer to a treatment would make the manuscript more easily readable. Possibly adding a table instead of description.
In the result section, major part of the manuscript is sequencing data, yet if the second largest phylum remains unassigned, the data should be rechecked. As it is possible to look into the abundance of methanotrophic bacteria, one should go to that level instead of phylum.
Discussion should berevised as about paddy fields and sraw addition many studies have been conducted. To describe the effect of chinese vedge and how we can mitigate methane or use less nirtogen fertilized could be the niche of this manuscript.


I have found several specific points whic improved, could benefit the manuscript.

Introduction:
L39-40 sentence lacks a verb. Suggestion: Instead of also use is
L51: changes over the cultivation period
L59-60: after the return alter to addition of Chinese milk vetch to paddy fields
L60: paddy soil or rhizosphere soil?
L60-61: Our goal in this study->The goal of this study or our goal; in this study is redundant. It is a long sentence with not clear message, e.g under the condition of straw. I suggest to restructure
L63-64: Which inputs? Microbial and fungal communities instead of components. As the title is about methane emissions, it would be good to add that to the importance.
General suggestion for the introduction:
As this manuscript addresses methane emission and microbial community changes, the authors could add a paragraph to explain how green fertilizer addition affects the methane emissions via microbial community changes. Explain that most of emitted methane originates from the methanogenic microorganisms present in the paddy field soil.

Materials and methods:

L71-73Authors could condense the lines by in average the annual sunshine was X, rainfall Y, etc. Instead of repeating average.
L74: How is yellow mud soil developed by river impact material. What is the river impact material?
L94: Check the formula for calcium magnesium phosphate
In the field experiment, nitrogen fertilizer was added to the experimental plots as well. How do the amounts of nitrogen in the form of urea compare to the amounts of fertilizer supplied to the fields where no vetch is cultivated over the winter?
L107: Could the authors explain, what is meant by irrigation management being the same as local routine? The reader very likely has no idea which is the local routine there.
L112-113: Could the authors explain the pesticide and herbicide local practices. How often, which ones and in which amounts were these added?
L115-117: Rice root microbiota is a different microbiome than the one in the rhizosphere and soil. Please use same term throughout the manuscript, especially for the samples you investigated.
L120: for microbial analyses->until further processing. Either DNA extraction or soil analysis.
L129: Add a space before reference
L150: Could the authors add the packages which were used in R for analysis?
Results:
L192: what is early rice ear stage?
L193-196: It is unclear what is meant under rice winter fallow, green manure rotation. Please rephrase.
In comparison to the control, the higher methane emissions when rice is planted, with or without straw could be expected. Could the authors elaborate more on the effect of the straw on methane emission or different nitrogen treatments?
L205-206: For clarification: in case the fertilizer was applied, treatment a, with or without straw addition, the rice yield was the highest?
L206:capital letter I the middle of the sentence
L213: space needed before reference
L225: Do authors mean rarefraction curves? What is meant by data was reasonable?
L235: How was the data analysis carried out, which resulted on the phylum level unassigned group to be second most abundant? Were these the unclassified reads pooled? On phylum level it is unlikely to have a large fraction of unassigned reads I these passed the quality control.

Reviewer 3 ·

Basic reporting

English
The English language could be improved in the text. More in detail:
- Line 91-92: substitute the original sentence with: “the two factors were combined to yield six treatments (i.e Va, Vb, Vc, SVa, SVb, and SVc). A control treatment (NSV) was established with winter fallow before rice cultivation”
- Line 95: substitute the original sentence “normal phosphate and potassium fertilizers” with “usual phosphate and potassium fertilization management”
- Line 132: substitute the original sentence with “as reported by Avnimelech et al., 2001.”
- Line 157: substitute with “air-dried and sieved at 6 mm size.”
- Lines 305-309: the structure of this sentence should be improved. You could divide in two separates sentences.

Literature references, field background/context.
Your introduction provides the right context and background for many of the points. However, it needs to be improved for the following general aspects and specific points:
- Lines 33-40: the reasons and the strategies for the reduction in the use of chemical fertilizers, for the increase in crop NUE and for the use of green manure and straw return to the soil should be presented in a more comprehensive way, more related to the experiment carried out and bringing more references from the literature. More in detail, the relationship between the above-mentioned themes and the expected effects on soil fertility properties, soil and rhizosphere microbial and fungal population, and greenhouse gases and methane emissions, especially for paddy systems, should be described. You may include the following references: Dawson, et al.. "Characterizing nitrogen use efficiency in natural and agricultural ecosystems to improve the performance of cereal crops in low-input and organic agricultural systems." Field Crops Research 107.2 (2008): 89-101; Bhattacharyya, P., et al. "Effects of rice straw and nitrogen fertilization on greenhouse gas emissions and carbon storage in tropical flooded soil planted with rice." Soil and Tillage research 124 (2012): 119-130; Bertora, Chiara, et al. "Dissolved organic carbon cycling, methane emissions and related microbial populations in temperate rice paddies with contrasting straw and water management." Agriculture, ecosystems & environment 265 (2018): 292-306.
- Lines 115-117: move this sentence to the introduction section.
- Lines 44-48: you should provide references that could explain why the experiment focused on microbial and fungal population in the rhizosphere and not also soil, the microorganisms investigated and which are the expected effects for plant nutrition and methane emissions due to microorganisms structure change. For reference, also see Conrad, R. (2007). Microbial ecology of methanogens and methanotrophs. Advances in agronomy, 96, 1-63; BERG, Gabriele, et al. Unraveling the plant microbiome: looking back and future perspectives. Frontiers in microbiology, 2014, 5: 148.
- The scientific context (and reference) for the methane pot experiment is not provided, as well as the illustration of available methodologies used in this field.

Article structure, figures, tables. Raw data.
Specific comments on Tables, Figures and Raw data are:
- Table 2: the comparison of CH4 emissions in the different treatments are well provided in Figure 2, as emissions rates. For this reason, it is better to remove this parameter from the table and to include the results in terms of kg/ha in the text, comparing this data with data from literature.
- Table 3 should be structured as Table 2, with parameters in column and treatments in rows. Also consider to improve the visualization of the table by separating bacterial and fungi data.
- Figure 1 shows different p(F) values. However, the ANOVA should be performed as for Table 2 data;
- Figure 1 shows the pot rice production data with different findings compared with field data, which causes confusion and are not discussed in the paper. You should explain the findings from the pots experiment in the text and maybe removed them from the figure, while providing them as supplemental materials;
- Figure 1: It is not clear if the yield data refer to one rice variety or if an average of the two varieties. This aspect is very important, if rice varieties show a similar or different effect.
- Figure 3 should be simplified, for helping readers focusing on the main aspects. Graph A to D, that shows results not discussed in the text, could be removed from this figure and included in supplemental materials.
- Raw data for Table 2 should provide the replication numbers for each treatment

Hypotheses and results.
- The hypothesis formulation should be improved in the introduction section to be coherent with the results presentation in the tables and figures (e.g. mid-term effects on: rice production, soil properties, microbial and fungi structure and diversity, methane emissions).

Experimental design

The introduction section should be improved for providing a broader context for a wider audience in biological and environmental sciences (see also specific comments).

Research question and knowledge gap.
The objectives of the experiments are stated in Lines 60-65 of the paper. You should improve the clarity of the paragraph and include the following aspects highly relevant for the experiment meaning: 1- the effects are studied after four years of treatments (rotation) establishment (Line 83: field experiment started in winter 2015, Line 117: soil collection in June 2019); 2 – if also rice varieties effects are investigated or not (line 106-107); 3 - treatments also refer to different fertilizer strategies for reducing total N supply; 4 - the research question also includes the effect of treatments on rice yield and methane production.
Moreover:
- Line 61-63: according with Table 2, you should also state that “the changes on soil properties” due to treatments are evaluated, not only the relationship with microbial and fungal changes.
- Line 62: it is not clearly stated which parameters are “environmental factors”, which seem to only refer to soil parameters.
- Lines 64-65: it is not clear if “mechanisms” refer to microbial or chemical driven effects.

Rigorous investigation performed to a high technical & ethical standard.
Following main methodological aspects should be better explained:
- it is not clear how the two rice varieties were treated in the experiment in the soil collection and results analysis. The effect of the different rice varieties is very important and should be presented if available.
- the pot experiment should present in a clearer way for the following aspects: Lines 152-155: It is a parallel field incubation experiment? Four pots for each treatment were incubated in a field closed to the main field experiment, and the results of the four pots were merged for having one replication? Lines 158-161: it is not clear if you placed the plastic pots in the soil or on the surface and how you added water to the dry soil to reach the water conditions for the experiment and how these conditions were maintained during the experiment. Possibly add a reference for this experiment.
- The soil properties before the experiment (Lines 74-77) should be included in the experiment and discussed or, alternatively, only a brief description could be provided in the “site description” section, in order not to generate possible confusion on the experiment’s objective.

Methods described with sufficient detail & information to replicate.
Materials and methods section could be improved:
- Lines 73-74: A better description of the soil profile should be provided, using an international classification and/or literature reference. Moreover, soil texture data are lacking.
- Lines 78-80: the data on vetch should be provided in the “ field experiment” section, with more information on vetch cultivation and soil incorporation technique, methodology utilized for estimating C and N supplied, the biomass yield if available, year-repetitions if measured .
- Lines 100-101: the plant stage of the second top-dress fertilization is not clear (i.e booting or heading stage?). Please use international standard for definition and description, as BBCH identification keys.
- Lines 105-106: you should describe sample collection (the area of harvest) and samples management (e.g dry matter determination, etc.).
- Lines 106-107: the presence of two different rice varieties and growing cycle durations should be presented at the beginning of the paragraph.
- Line 172: a briefly description of the methodology for calculating methane emissions should be provided
- Line 174: add a reference to this sentence from a similar agro-environment
- Lines 192-193: Was the rotation maintained from the establishment of the experiment (winter 2015) to soil and rhizosphere sampling (June 2019) at rice heading stage? If this is the case, you should emphasize this aspect (i.e mid-term effect of rotation/treatments)

Validity of the findings

The presence and data collection and analysis of two different varieties must be clarify.

The results section and related disccusions needs to be improved:
- Lines 193-202: the presentation in the text of table 2 results, which are high relevant for the experiment, should be expanded. For instance, which treatments are statistically different? Moreover, methane results should be separated from soil properties results.
- Lines 205-206: Figure 1 results should be presented in a clearer way (see comments on Figure 1);
- Lines 206-208, lines 212-216 and lines 219-220: these sentences should be moved to the discussion section.

Additional comments

The paper needs a deep improvement, especially in the following points:
- how you include the presence of the two rice varieties in the experimental design;
- the pot experiment should be presented as a further parallel investigation;
- The results section should be expanded (see specific comments).
- The discussion section should be improved. It is well structured as paragraph division, but you should illustrate the connection between your experiment results and what you are discussing in the text. More numeric data, examples and comparisons with the results from the literature should be presented in this section, before conclusive statements.
- Tables and Figures should be simplified as illustrated in specific comments of this revision

---

## Round 0.2 · Major Revisions

The authors incorporated most of the suggestions and revised the manuscript. However, the revised version still needs several corrections, such as language still needing help, the statistical analysis also needs some clarifications and other major points as raised by reviewers (as mentioned below in the reviewers' comments). Please revise your manuscript accordingly.

·

Basic reporting

Basic reporting
There are still some problems with English and many typos throughout the manuscript that make it hard to follow sometimes, however I suspect PeerJ's language editing service to be of great help here. My suggestions have been addressed in part and thus have some followup issues.

Discussion
(Line 288). Sentence seems incomplete.
(Line 290). Unclear what is being said in this sentence.
(Line 291). This conclusion, in whole, doesn’t seem to be supported by the data in Figure 2. Yes, the CH4 emission were highest in SVa which had high levels of CH4 emissions compared to the other treatments, but the others are all similar. Thus, how can you say that it is related to N input? If it were, you should see a similar effect in the Va-c treatments, where Va would be the highest CH4 emissions relative to Vb-c. I would change this statement so that it better reflects the data.
(Line 303) Space between Methanomicrbia and after. Sentence also unclear.
(Line 305) Capitalization of long-term.
(Line 299). This whole paragraph should be given a better look for English corrections. It is quite hard to follow. Also, there are many statements about microbial abundances higher in a certain experimental group without statistical support. This needs to be included.
(Line 302). It is quite difficult to see the % differences of Methanomicrobia in this plot with the current colors. If emphasizing a particular group of microbe, perhaps it is better to show that group’s abundance in a separate plot across the treatments being compared.
(Line 305). Sentence incomplete.


Conclusions (line 325): The first sentence is more clear when it is apparent to the reader that the statement is in relation to the control condition.
(Line 328) When abundance comes into play, indicating that these are relative numbers is important. Thus, the relative number of methanogens may have increased (or specific methanogen clades) but it is unclear from the current data whether the total number was affected. In addition, there is insufficient evidence from the current study to suggest any of these organisms is 'beneficial.' I suggest more cautionary language such as 'punitively beneficial' or remove 'beneficial microorganisms' entirely.

Experimental design

No comment.

Validity of the findings

I have commented above on some statements that are not justified given the data and would need to be adjusted to better fit the data.

Additional comments

Thank you for addressing my previous concerns. There are a few followups that I feel are necessary before the manuscript is suitable for publication. I feel there are certain areas that can be clarified in order to match the statements and conclusions made and what the data supports. This is particularly true in the discussion and conclusion sections.

Reviewer 3 ·

Basic reporting

General comment:
- The use of the English language is not completely adequate in several sentences throughout the text.
Specific comments:
- line 14: add “emissions” after “the risk of greenhouse gases”
- lines 14 and 15: substitute the original sentence with “…the presence of Chinese Vetch and the management of rice straw and nitrogen fertilization….”
- lines 16 and 17: substitute the original sentence with “We investigated methane emissions, abundance and diversity of the microbial community in the rhizosphere after incorporation of …”
- line 22: substitute “The incorporated” with “The soil incorporation”
- the original sentence in line 25 and 26 should be inserted in line 23 and modified in: “Methane fluxes are also affected by nitrogen application.”
- Lines 35 and 36: substitute the original sentence with: “one of the main topics studied by scientists working in the related fields worldwide”
- Line 74: “in the present study” instead of “at the present study”
- Line 98: substitute “in the growing season” with “during the growing season”
- Lines 119 and 120: the original sentence should be substitute with “Fresh Chinese milk vetch at the time of incorporation showed the following characteristics:”
- Line 124: Substitute “in” with “at”
- Line 142: “as reported” is written twice, correct
- Line 145: add “In order to…” at the beginning of the sentence
- Line 217: substitute “in the heading…” with “at the heading…”
- Lines 223-224: substitute the original sentence with “the soil bulk density in the treatment of Chinese milk vetch and straw incorporation decreased significantly…”
- Line 250: “abundance” instead of “abundant”

General comment:
- Although showing the context, the introduction does not adequately present the state of the art for some of the issues related with the experiment. For instance, are there studies about NUE or methane emissions assessment after green manure and straw in rice systems? Which questions are still opened? In general, sentences seem to be not well connected one to another as it is difficult to follow the rationale which leads to the objectives.

Specific comments:
- Lines 36 and 37: the sentence is not completely developed and its relevance is not clear, as breeding is not treated in this paper;
- Lines 37 and 38: reformulate the sentence: “green manure and straw management represent agronomic strategies for…”

Specific comments on Tables and Figures:
- Table 2 and Table 3: In my opinion, the results could be better visualized if data of each measured parameters would be in the same column.
- Figure 2: You should add in the caption that reported data refer to the average of n measurements and specify the interval of measurements.
- A full presentation of ANOVA results tables is needed.

Experimental design

General comments:
- One of the main problems is that, in line 91-92, it is stated that “The experiment was a split plot”, with “Straw returned to soil as main plot” and N levels as split plot. If this is the case, how NSV treatment is included in the design as it has only one level?

Specific comments:
-The objectives of the experiments are stated in Lines 74-77 of the paper. The sentence should be improved: green manure and rice straw effects are also evaluated, not only “fertilizers strategies”. For instance, you could substitute the original sentence, with the following one: “In the present study, we studied the effects of different fertilization and rice straw management in Chinese vetch-rice rotation after four years. We investigated (1) changes in soil properties, (2) rice yield and straw production, (3) methane emissions (dynamics?) during rice growth and (d) rhizosphere bacterial and fungal communities and their relationships with soil parameters.
- The methodologies utilized for soil properties characterization are not fully presented in the correspondent paragraph (lines 134-142). For instance, lack of methodology illustration for AOC, MBC and MBN listed in Table 2;
- Lines 98 and 99: It is not clear that the total amount of N is the same in all the treatments but with a different fractionation between vetch and rice as reported in Table 1;
- Lines 110 and 111: it is not clear why late rice management is illustrated. It is also included in the experiment?
- Line 126: if it is correct, insert “After 4 year of treatment application,…”

Validity of the findings

General comment:
- The results of the methane flux dynamics during rice growth should be presented and discussed in the paper (not only the average of the period)

Specific comment:
- Line 227: check the value of the reported yield

Additional comments

The paper needs a deep improvement especially for the points specified in the "general comments" of the above sections of the revision.

---

## Round 0.3 · accepted · Accept

The authors amended the manuscript as per comments and suggestions, and also corrected the language. The revised version is much clearer and reads better than the previous version, and could be accepted in the current revised version.